# Change in the Results of Motor Coordination and Handgrip Strength Depending on Age and Body Position—An Observational Study of Stroke Patients and Healthy Volunteers

**DOI:** 10.3390/ijerph19084703

**Published:** 2022-04-13

**Authors:** Anna Olczak, Aleksandra Truszczyńska-Baszak, Józef Mróz

**Affiliations:** 1Military Institute of Medicine, Rehabilitation Clinic, 128 Szaserów Street, 04-141 Warsaw, Poland; jmroz@wim.mil.pl; 2Faculty of Rehabilitation, Józef Piłsudski University of Physical Education in Warsaw, 00-968 Warsaw, Poland; aleksandra.truszczynska@awf.edu.pl

**Keywords:** stroke, older people, age ranges, grip strength, motor coordination, stabilization

## Abstract

Objective: The stroke is considered a common disease of the elderly. Young people also get sick, but the risk of stroke increases with the age of 60. Stroke, regardless of the age of the patients, causes functional deficits; therefore, the aim of the study was to analyze the significance of the body position and examined upper limb on the parameters of motor coordination and handgrip strength in various age groups of people after stroke and healthy people. Material and method: This is an observational study. A total of 117 people participated in the study (60 stroke patients and 57 healthy people without neurological disorders). Both patients and healthy volunteers were prospectively divided into three age groups: 18–45, 46–60, and 61+. The tests were carried out in two starting positions: sitting without back support and lying on the back with the upper limb stabilized against the body. HandTutor^TM^ and a hand dynamometer were used to assess the motor coordination, including the maximum range of motion and frequency of movement, as well as the grip strength. Results: The passive stabilization of the trunk and shoulder improved the maximum wrist ROM (*p* < 0.001) and frequency of finger movements (Hz F5 *p* = 0.018; F3 *p* = 0.010; F2 *p* = 0.011), especially in the oldest stroke patients. In the group of healthy volunteers, the most statistically significant results were obtained in the age range of 46–60. They occurred in both stable (wrist maxROM *p* = 0.041 and Hz F5 *p* = 0.034; Hz F4 *p* = 0.010; Hz F3 *p* = 0.028; Hz F1 *p* = 0.034, maxROM F1 *p* = 0.041) and unstable positions (maxROM F5 *p* = 0.034; maxROM F4 *p* = 0.050; maxROM F3 *p* = 0.002; maxROM F2 *p* = 0.002). In the group of the oldest healthy people, only one significant result was obtained in the stable position (Hz F3 *p* = 0.043). Conclusion: Passive stabilization of the trunk and examined upper limb improves the results of motor coordination of the distal part of the upper limb in both study groups. Passive stabilization of the trunk and upper limb improves motor coordination, especially in the oldest group of patients, after stroke.

## 1. Introduction

Stroke is the third most common cause of death in developed countries (after cardiovascular diseases and cancer) [1,2,3]. It is quite commonly believed that stroke is a common disease in the elderly. In reality, however, young people also suffer from them; however, the causes of stroke differ significantly from those assigned to adulthood. Regarding the statistics for young people, there is a significantly higher number of cases among men. The proportions even out with age, and all differences virtually blur with age [4,5,6].

People all over the world are living longer. The world population aged 60 and over is projected to be 2 billion by 2050, compared with 900 million in 2015. Today, 125 million people are 80 years or older. By 2050, there will be 434 million people in this age group worldwide [7,8].

Age changes begin after the age of 40. The World Health Organization characterizes the age ranges, according to which, from the age of 18 to 45, we can observe peak physical fitness; however, after the age of 40, we can observe a slow decline in fitness and physical strength. Some people over forty experience a lack of energy, fatigue, and weakness, as well as a decrease in muscle mass and strength and motor coordination of 10% [8,9].

After the age of 60, we observe a further decline in life forces, physical fitness, and biological functions. Additionally, after the age of 60, we are talking about three periods of old age. According to the World Health Organization, these are early old age (from 60 to 75 years old), late old age (from 75 to 90 years old), and old age or longevity (over 90 years of age). These periods are characterized by a further decrease in physical and mental fitness; although, the level in each of these periods depends on the biological condition of a person, congenital or past diseases, accidents, etc. [8,9]. 

The risk of having a stroke increases as you reach the age of 60. At this age, atherosclerotic changes are extremely dangerous and lead to the narrowing and obstruction of the arteries; the course of a stroke and its effects are usually much more serious than in the case of young people [4].

Of the post-stroke patients, almost half reported a deterioration in the function of the upper limb and hand [10,11]. It usually takes longer for the upper limb to recover than the lower limb. This may be due to too short a unit therapy time for the upper limb and/or hand therapy. Hayward et al. noted that “arm training lasted from 4 to 5.7 min and from 23 to 32 repetitions per session during physical therapy, also in a hospital setting and slightly longer, from 11 to a maximum of 17 min per session during occupational therapy” [12]. Perhaps the process of improving the upper limb alone is not sufficient, not to say inadequate. Researchers prove that, after a stroke, the level of bilateral motor coordination decreases; therefore, they devote a lot of attention to two-hand coordination, considering it to be crucial in post-stroke patients [13,14]. Therefore, most reports on the motor coordination of the upper limb present the results of two-handed coordination in patients after stroke [15,16,17,18,19,20]. 

In our work, we assessed motor coordination and handgrip strength in patients after stroke in various age groups. We applied the results of the research to healthy volunteers, taking into account the same age ranges as in people with neurological deficits. However, an important part of the research was the assessment of the coordination and strength in two different positions of the torso and examined upper limb. The authors had carried out similar studies earlier, and the results of these studies indicated that the lying position with the stabilized upper limb against the body of the subject is important in reconstructing the motor coordination of the distal part of the upper limb [21,22]. Similar studies, showing the importance of body and upper limb position on the possibility of triggering movement in the paretic upper limb, were presented by Souque in 1916 and others, such as Brunnstrom (in 1970) and Nijland et al. (in 2010), who reported that two simple tests (finger extension and arm abduction) could be used to assess the possibility of restoring function in the upper limb [23,24,25].

Researchers also assessed the conditions for achieving greater grip strength in both stroke patients and healthy volunteers. They proved that the position of the forearm was important for obtaining more grip strength: “The best parameters were recorded in the transverse position and in the lateral plane compared to the plane coinciding with the body axis and horizontal position” [26]. Okunribidi et al., in turn, investigated how the stabilization of the forearm affects the strength results in healthy people and proved that a stable position is especially important for obtaining a greater grip strength [27]. In turn, Kachanathu et al. investigated the effect of shoulder stabilization exercises on isometric handgrip strength in patients with unilateral shoulder syndrome, and a significant difference was found after the use of stabilizing exercises [28]. The aim of the study by Khanafer et al. (2018) and other researchers was to assess the “age-related differences in compensatory shoulder-trunk coordination when reaching with trunk support”. The tasks were carried out with eyes closed. Participants performed two tasks with dominant and non-dominant arms, both at high and preferred speeds. Younger and older participants achieved similar results, with the difference that the results in the elderly were much more variable and inconsistent than in the younger adults [29,30]. 

In turn, Na Kyung Lee et al. proved that sensorimotor processing, including the motor skills of the dominant and non-dominant hands, deteriorates during the aging process [31].

So, the researchers proved that coordination worsens in older adults. 

It is difficult to clearly present what is important for the improvement of the motor coordination of the upper limb and whether there is a chance for improvement of motor coordination in the elderly and neurologically disturbed, who suffer from stroke more often.

Therefore, the aim of the study was to analyze the importance of the position of the body and examined upper limb on the parameters of movement coordination and handgrip strength in various age groups of people after a stroke and healthy individuals.

## 2. Material and Methods

### 2.1. Study Design

This is an observational study. The aim of this experiment was to analyze the parameters of motor coordination and handgrip strength in stroke patients and healthy volunteers in various age groups, as well as the selected trunk and upper limb positions. Both patients and healthy volunteers were prospectively divided into three age groups: 18–45, 46–60, and 61+. Thus, the maximum range of motion (max ROM), frequency of wrist (Hz) and finger (F) movements, and grip strength (dependent variables) were valued in three age groups and different starting positions (independent variables).

### 2.2. Ethical Approval

The study was carried out in the Teaching Department of Rehabilitation of the Military Medical Institute (MMI) in Warsaw, Poland. It was approved by and carried out in accordance with the recommendations of the Ethical Committee of the Military Medical Institute (MMI; approval number 4/MMI/2020). Prior to inclusion, all subjects were informed about the purpose of the study. Written informed consent was obtained from all subjects, in accordance with the tenets of the Declaration of Helsinki.

### 2.3. Subjects

Recruitment of patients, according to the defined inclusion/exclusion criteria, consisted of the assessment of the patient by a physiotherapist (AO) participating in the studies (with tests/scales: TCT, FMA-UE, and MAS), after examination by a medical doctor (JM) admitting patients to the clinic. Healthy volunteers (after verification, according to the exclusion criteria) were also assessed with selected tests/scales.

A total of 180 people were examined before inclusion: 63 people (15 stroke patients and 48 healthy) were excluded because of their functional condition and declined to participate. Finally, 117 males and females were recruited from among patients of the Teaching Department of Rehabilitation of the Military Medical Institute (MMI) and University of Social Sciences in Warsaw, including 60 post-stroke patients (study group) and 57 neurologically undisturbed volunteers (control group). The flow of participants, through each stage of the study and for analysis, is shown below (Figure 1).

The group of post-stroke patients were 4–7 weeks past stroke of the disease (unilateral subcortical), with stable trunk (the trunk control test 74–100 points), subjects were in a functional state, allowing for movements of the upper extremity (FMA-UE 43–49 motor function points and normal sensation/light touch); the tension of forearm and hand muscles were measured with modified Ashworth scale (MAS 1/1+) [32,33,34]. There were neither biometric differences nor tests to qualify a stroke survivor between different age groups of patients (Appendix A).

The control group (57 people) consisted of healthy, neurologically undisturbed volunteers (trunk control test (100 points) (FMA-UE 66 motor function points)). The tension of forearm and hand muscles were measured with the modified Ashworth scale (MAS 0) [33,34,35]. The group of patients after stroke and healthy volunteers did not differ, in terms of biometric data, in the age ranges studied. The characteristics of the subjects are shown in Table 1 and Table 2 and the Appendix A.

Stroke group inclusion criteria: (1) patients with ischemic stroke (unilateral subcortical); (2) trunk control test 70–100 points; (3) subjects who were in a state allowing for movements of the upper extremity (FMA-UE 40–66 motor function points); (4) muscle tension (MAS 0/1/1+); (5) no severe deficits in communication, memory, or understanding; and (6) at least 18 years of age.

Stroke group exclusion criteria: (1) stroke up to four weeks after the episode; (2) another neurological disease entity; (3) lack of trunk stability; (4) no wrist and hand movement; (5) muscle tension (>2 MAS); (6) high or very low blood pressure; (7) severe deficits in communication, memory, or understanding; and (8) dizziness and the malaise of the respondents. 

Control group inclusion criteria: (1) the control group consisted of healthy subjects, free from the upper extremity motor coordination disorders; (2) at least 18 years of age.

Control group exclusion criteria: (1) a history of neurologic or musculoskeletal disorders, such as carpal tunnel syndrome, tendonitis, stroke, head injury, or other conditions, that could affect their ability to active movement and handgrip; and (2) severe deficits in communication, memory, or understanding. 

### 2.4. Motor Coordination Assessment

The HandTutor^TM^ device (MediTouch, Netanya, Israel) and electronic manual dynamometer EH 101 (Camry, Shiqi, China) were used for handgrip strength measurement (error of measurement, 0.5 kg/1 lb).

HandTutor^TM^ allows for measurements of the frequency (i.e., the number of cycles per second, where one cycle represents the movement from flexion to contraction) and maximum range of movement, which is automatically measured during the frequency test, which were performed over time (10 s) (sensitivity: 0,05 mm of wrist and fingers ext./flex; frequency of movement (motion capture speed: up to 1 m/s)) [21,22]. The system is based on a safe and comfortable glove, equipped with position and motion sensors (sensitive electro–optical sensors evaluating the position and speed of wrist and finger movement; power supply: voltage: 5 V dc, rated current input: 300 mA) and the Medi Tutor TM software. Medi Tutor presents data in Excel. The maximum range of motion data during fast frequency estimation movements is represented as ROM [36]. 

Before each test, the patient was instructed on how the exercise should be done. The test consisted of two motor tasks, carried out in two different starting positions: sitting and lying down (supine). During the first examination, the subject sat on the therapeutic table (without back support), with their feet resting on the floor. The upper limb was to be examined in adduction, with the elbow bent in the intermediate position between pronation and supination of the forearm. In the supine position, the upper limb was stabilized at the subject’s body (adduction in the humeral joint, elbow flexion in the intermediate position). 

In each of the starting positions, after putting the glove on, the subject was asked to make moves as quickly and in as full a range as possible. Finally, the measurement of grip strength, with a dynamometer, was performed in both analyzed starting positions, after completing the range of motion and speed or frequency tests. The pause time between the coordination and handgrip strength measurements was 15 to 20 min and resulted from the procedure of preparing the subjects for the test. The upper extremity tested in the stroke patients was the paretic extremity. In healthy subjects, the dominant hand was tested. 

### 2.5. Sample Size Calculation

The sample size was calculated using the G * Power 3.1.9.4 program. It was calculated based on the assumption of the Wilcoxon test, in order to compare the results between items. A strong effect of d = 0.8, α = 0.05, and test power of 0.8 were assumed. At that time, the minimum number of people was 12 for the studied groups.

### 2.6. Statistical Analysis

Statistical analyses were performed using IBM SPSS Statistics 25.0. In order to compare the two groups, in terms of the analyzed parameters, the analyses were carried out with the Mann–Whitney U test; when there were more compared groups, analyses were performed with the H Kruskal–Wallis test. For the post hoc analysis in the Kruskal–Wallis test, the Dunn test, with a correction of the Bonferroni significance level (correction for multiple comparisons), was used. The Wilcoxon test was used to compare the two measurements. The level of significance was α = 0.05.

## 3. Results

### 3.1. Comparison of Age Groups among Patients after Stroke

In order to compare the three age groups, in terms of the analyzed parameters, an analysis was carried out using the H Kruskal–Wallis test. The analysis showed significant differences between the groups, in terms of: (a) position without stabilization (Hz wrist (cyc/s), HzF5, and MaxROM for F5, HzF4, HzF3, HzF2, HzF1, and max ROM for F1); (b) stabilized position (MaxROM for F1). In order to establish the nature of the differences between the age groups, post hoc analysis was performed using Dunn’s test, with correction of the Bonferroni significance level. Detailed analysis showed that people aged 61 and over obtained a significantly lower result for Hz wrist (cyc/s), HzF5, and MaxROM for F5, HzF4, HzF3, HzF2, and HzF1; and subjects aged 46–60 years had lower results for max ROM for F1 and Hz wrist (cyc/s) than people aged 18–45 in the measurement of the position without stabilization. For the parameters in the stabilized position, after adjusting for the significance level, the differences between the age groups turned out to be statistically insignificant. The results of the analyses are presented in Table 3.

Then, using the Wilcoxon test, the results of the measurements in the non-stabilized and stabilized positions in individual age groups were compared.

The analysis showed no significant results in the 18–45-year-old group. (Table 4). On the other hand, among the subjects aged 46–60 (Table 5), in the stabilized position, a higher result was obtained for MaxROM F4 and F1.

For the oldest group (Table 6), in the stabilized position, significantly higher results were obtained for MaxROM Wrist, HzF5, 3, and 2, with lower results for MaxROM F4 and F3, than in the position without stabilization. 

### 3.2. Comparison of Age Groups among Healthy Patients

In order to compare the three age groups among healthy people, in terms of the analyzed parameters, the H Kruskal–Wallis test was performed. The analysis showed significant differences between the groups, in terms of Hz F1 in the non-stabilized and stabilized positions for the Hz of the wrist. In order to establish the nature of the differences between the age groups, a post hoc analysis was performed using the Dunn test, with a correction of the Bonferroni significance level. Detailed analysis showed that, in the position without stabilization, people aged 18–45 years obtained a significantly higher result for Hz F1 than those aged 46–60 years (*p* = 0.043). In the stabilized position, people aged 46–60 years obtained a significantly higher result for Hz wrist (cycle/s) than people in the oldest age group. The results of the analyses are presented in Table 7.

Then, using the Wilcoxon test, the results of the measurements of healthy people in unstable and stabilized positions were compared in individual age groups.

The analysis showed that, in the 18–45 age group in the stabilized position, higher results were obtained for the MaxROM of the wrist than in the non-stabilized position (Table 8).

Among the subjects aged 46–60 years (Table 9) in a stabilized position, higher results were obtained for Max ROM of the wrist, Hz F5, Hz F4, Hz F3, and Hz F1, with lower results obtained for MaxROM F5, MaxROM F4, MaxROM F3, MaxROM F2, and MaxROM F1, than in the position without stabilization.

One significant difference was noted for the oldest group (Table 10)—in the stabilized position, the result was higher for Hz F3 than in the position without stabilization. 

## 4. Discussion

The results of the study showed that the greatest improvement in wrist and hand coordination in post-stroke patients was achieved in the supine position, with a stabilized upper limb. Moreover, the results favor the oldest group of patients. 

Motor coordination was assessed using the HandTutor^TM^. This device has already been used to test the ranges of passive and active mobility, as well as motor coordination and the maximum ranges of motion during the assessment of the frequency of wrist and hand movements [21,22]. It also appeared in the work of Carmela et al.; although, in their work, the device, apart from the test function, was assessed in terms of the effects of therapy [37]. For the functional assessment of the subjects, commonly accepted scales and tests, i.e., the trunk control test and Fugl-Meyer score, as well as the modified Ashworth scale, were used [32,33,35].

In our study, we divided into age groups: 18–45, 46–60, and 60+. We were guided by the WHO information and assessments of psychologists; however, most of all the division and size of the groups depended on what patients and healthy volunteers we could recruit. In addition, for young adults, according to life cycle psychologists, most people reach adulthood between the ages of 19 and 39. Erik Erikson writes that people reach this point at different ages, due to many factors; according to him, there is no absolute schedule for young adulthood; however, most often, this age range is between 18 and 35 years old. Meanwhile, middle-aged adults are people aged 35 to 55 or 65 [4,9,38]. On the other hand, in the fields of health, medicine, and human development, young adulthood is the time when people are traditionally the healthiest. This is the stage between adolescence and adulthood, which falls roughly between the ages of 15 and 29, with the average ages of 30–55 and 60; old age would be considered to be after 60 [4,9,38].

The results of our research show a certain tendency to change the parameters of motor coordination, depending on the body position, age of the respondents, and burden of the disease or its absence. In the group of people after a stroke, sitting in an unstable position, the results became significantly higher as the age of the examined patients decreased. In the smallest range of the youngest patients, the results were significantly the highest; in the group of the oldest patients, the results were the lowest. On the other hand, in the stable position, there were no significant differences between the age groups. The analysis of the influence of the test position on the results of coordination and grip strength showed that the position did not matter for the group of the youngest patients. In the 46- to 60-year-old group, significant results were recorded in a stable position only for Max ROM F4 and F1. On the other hand, in the group of the oldest patients, the stabilization of the trunk and examined upper limb resulted in a significant increase in parameters for several variables: max wrist ROM, max ROM F3, F4, Hz F5, F3, and F2.

In the group of healthy volunteers, the results were significantly higher in both the stable and unstable positions, but they concerned only the two Hz F1 results in the unstable position and Hz wrist in the stable position. Here, too, the youngest achieved higher results in both examined positions. For the youngest group of healthy volunteers, the stable position was only more statistically significant in the case of max wrist ROM. The most significant results were achieved in the age group of 46–60 years. These were for max wrist ROM and all fingers, as well as for finger Hz, except Hz F2, but these results occurred in both stable and unstable positions. Among the oldest participants, there was one significant difference in the stable position, and it concerned Hz F3.

In conclusion, a stable position seems to be very important, especially for the group of the oldest stroke patients. In the group of healthy people in a stable position, significant results were recorded in all age groups; however, in the group of the oldest healthy volunteers, there were significantly fewer of them than in patients after stroke.

The study by Linares et al. (2013) investigated the influence of age on several attributes of sensorimotor performance during the running task. As in our research, the respondents were divided into three groups, according to age: group 1 (20–40 years old), group 2 (age 41–60), and group 3 (age 61–80). The Kruskal–Wallis test showed significant differences (*p* < 0.05) between the groups, with the exception of the variables of postural velocity in the dominant arm, posture speed, and initial deviation in the non-dominant arm (*p* < 0.05) [39]. The results of the cited study suggest that age introduces significant differences in the motor function of the upper limb. In our study, the age of stroke patients also played a role in the results achieved. What is more, in our study, we found that the stable position of the trunk and examined upper limb is important for significantly higher results of motor coordination. Analysis of the handgrip strength test results showed that neither age nor position influenced the results. In the cited study, the results show that there are objective differences in the sensorimotor function due to age, and the differences are greater in the case of the dominant arm [39]. On the other hand, in another study, the analysis of the stable and unstable position for the examination of the affected dominant and non-dominant hand of the patient after a stroke showed that the stable position is more beneficial for achieving better motor coordination parameters, especially in the case of the non-dominant limb [40]. In our current study, the dominant hand was tested 100% in a group of healthy volunteers, and they achieved higher results than in patients. After a stroke, the dominant limb was tested in 50% of the cases, as was the non-dominant one. Moreover, in the group of the oldest patients, more than half of the examined limbs were not dominant. Taking this into account, the study highlights the importance of a stable position for the oldest stroke patient group.

Our study, in three different age groups of stroke survivors and healthy people, as well as different starting positions, showed no effect on handgrip strength.

Kumar et al. identified factors that can affect the speed and coordination of the movement of the upper limbs. Healthy young people were examined, and a statistically significant difference between the groups was found, which suggests that smoking had a negative effect on the speed and coordination of the upper limb [41]. Moreover, the authors pay attention to the fragility syndrome of the elderly, as well as the fact that atherosclerosis very often affects not only the vessels in the brain [42,43]. These types of changes may translate into the deterioration of the fitness of people over 60 years of age.

In our work, we established the circumstances of improving the frequency of upper limb movements, as well as the age range for which a stable position is important.

Our work shows that passive stabilization of the trunk and upper limb, with reduced tension in patients after stroke, may affect the parameters of motor coordination of the distal part of the upper limb, and it is especially important for patients in early old age.

### 4.1. Research Value

Our research shows that the use of a stable position makes a special sense, when we work to improve coordination and handgrip strength in the “weaker” patients, such as older, post-stroke patients. Placing the patient in the supine position with a stabilized upper limb and paresis close to the body may favor the restoration of normal movement patterns, especially in the elderly after a stroke. 

### 4.2. Study Limitation

The limitation of the study is the unequal number of respondents in different age groups. Especially, the youngest group is the least numerous. This is because there are fewer strokes among young people. Therefore, we tried to adjust the study groups, in terms of other parameters, such as the distribution of the number of people in age groups, as well as the gender, body weight, or height of the respondents. Moreover, the limitation of the study is that the patients were examined in a good functional condition, enabling movement (e.g., muscle tension MAS 1/1+); however, the aim of the study was to assess the coordination of movements and grip strength, so the functional status of patients should be characterized by the possibility of any movement of the wrist and fingers. In addition, the relatively short pause time (15 to 20 min) between the coordination and handgrip strength measurements can be a value that alters the results. Therefore, in subsequent studies, this time will be extended. One should also consider the issue of the fragility syndrome of the elderly, as well as the fact that the elderly often also have atherosclerotic lesions in other parts of the vascular bed. In our research, we did not consider such data, which may be of importance for the achieved results. The above limitations should be taken into account in subsequent studies.

## 5. Conclusions

Passive stabilization of the trunk and examined upper limb improves the results of motor coordination of the distal part of the upper limb. Passive stabilization of the trunk and upper limb improves motor coordination, especially in the oldest group of patients after stroke. It is important to repeat the tests in equal quantitative and homogeneous groups.

## Figures and Tables

**Figure 1 ijerph-19-04703-f001:**
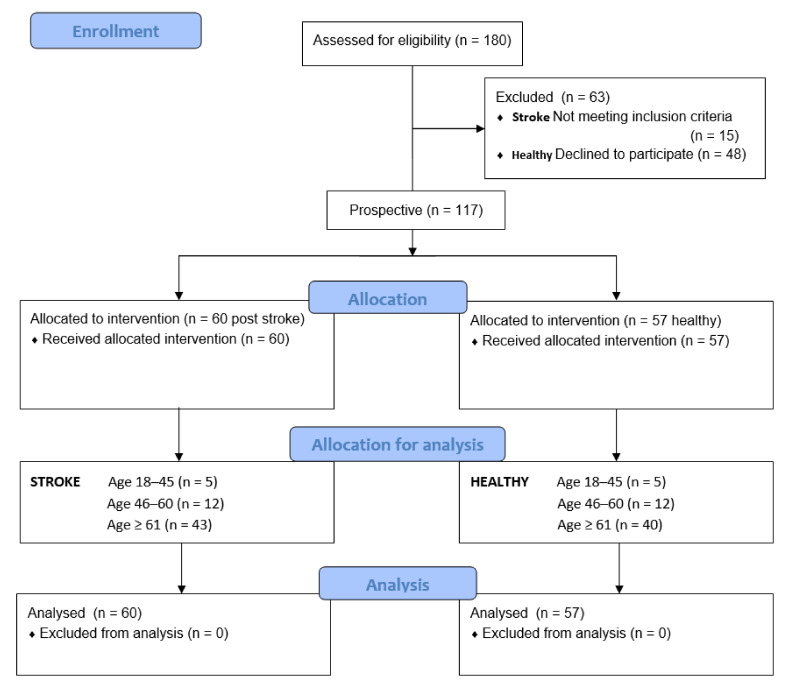
Flow of participants through each stage of the study.

**Table 1 ijerph-19-04703-t001:** Biometric data of the post-stroke patients and healthy control group.

Group	Stroke			Control		
	Height	Body Mass	BMI	Height	Body Mass	BMI
Age range 18–45	175.20 ± 9.88	71.20 ± 9.88	23.10 ± 1.38	173.80 ± 8.14	68.80 ± 11.08	22.63 ± 1.69
Age range 46–60	170.42 ± 5.68	75.50 ± 7.30	25.97 ± 1.82	167.17 ± 7.03	71.17 ± 11.34	25.56 ± 4.56
Age range > 61	170.67 ± 9.00	77.07 ± 10.16	26.40 ± 2.37	166.63 ± 7.80	73.75 ± 10.92	26.53 ± 3.25
H	1.03	1.40	9.89	1.14	3.42	9.33
*p*	0.597	0.498	0.007	0.564	0.181	0.009
Effect size	<0.01	<0.01	0.14	<0.01	0.02	0.13

**Table 2 ijerph-19-04703-t002:** The epidemiological data of post-stroke patients and healthy control groups.

Number of Participants *n* = 117 (100%)	Post-Stroke *n* = 60 (51.3%)			Healthy *n* = 57 (48.7%)		
**Age range, years** (M ± SD)	**18–45** (33.40 ± 8.88)	**46–60** (52.42 ± 4.52)	**>61** (72.56 ± 8.17)	**18–45** (31.80 ± 8.32)	**46–60** (49.75 ± 4.92)	**>61** (70.25 ± 6.76)
*n* (%)	5 (8%)	12 (20%)	43 (72%)	5 (8.8%)	12 (21%)	40 (70.2%)
Female	1	3	26	2	5	25
Male	4	9	17	3	7	15
Right-affected side	3	7	20	N/A	N/A	N/A
Left-affected side	2	5	23	N/A	N/A	N/A
Dominant hand (right)	5 (100%)	12 (100%)	43 (100%)	5 (100%)	12 (100%)	40 (100%)
Cerebral ischemic stroke unilateral subcortical (*n*/%)	5 (100%)	12 (100%)	43 (100%)	N/A	N/A	N/A
Time post-stroke/episode (week); subacute	4–7	4–7	4–7	N/A	N/A	N/A
Trunk control test (points 74–100)	89.80 ± 13.97	80.67 ± 11.66	82.74 ± 12.14	100.00 ± 0.00	100.00 ± 0.00	100.00 ± 0.00
FMA-UE (points 43–49)	46.20 ± 2.59	46.17 ± 2.08	45.67 ± 2.05	66.00 ± 0.00	66.00 ± 0.00	66.00 ± 0.00
MAS (degrees 0/1/1+) (examined n)	0/1/1+ 0/3/2	0/1/1+ 0/8/4	0/1/1+ 0/27/16	0/1/1+ 5/0/0	0/1/1+ 12/0/0	0/1/1+ 40/0/0

**Table 3 ijerph-19-04703-t003:** Comparison of age groups of patients after stroke, in terms of the parameters measured in the unstable and stabilized position.

Age Groups of Patients after Stroke	18–45 Years (*n* = 5)			46–60 Years (*n* = 12)			61 Years and Older (*n* = 43)						
	*M*	*Me*	*SD*	*M*	*Me*	*SD*	*M*	*Me*	*SD*	*H(2)*	*p*	*η* ^2^	*Post hoc*
Non-stabilized Hz wrist. (cyc/s)	2.16	2.40	0.51	1.15	0.90	0.90	1.03	0.90	0.60	8.33	**0.016**	0.11	1–3 *p* = 0.012
													1–2 *p* = 0.043
MaxROM (mm)	20.14	20.60	3.86	17.30	18.60	4.81	15.50	14.90	8.51	5.54	0.063	0.06	
Hz F5	2.40	2.40	0.60	1.58	1.60	0.94	1.39	1.10	0.87	6.41	**0.041**	0.08	1–3 *p* = 0.035
MaxROM F5	26.42	26.30	3.77	18.48	17.50	10.83	16.54	16.60	7.22	7.43	**0.024**	0.10	1–3 *p* = 0.019
Hz F4	2.40	2.40	0.60	1.58	1.60	0.94	1.37	1.10	0.88	6.41	**0.041**	0.08	1–3 *p* = 0.036
MaxROM F4	29.02	26.60	9.61	23.69	26.85	9.71	20.61	21.10	6.49	5.30	0.071	0.06	
Hz F3	2.40	2.40	0.60	1.58	1.60	0.94	1.37	1.10	0.87	6.63	**0.036**	0.08	1–3 *p* = 0.032
MaxROM F3	23.72	24.20	2.13	23.41	24.25	6.28	20.19	20.00	5.29	4.93	0.085	0.05	
Hz F2	2.40	2.40	0.60	1.58	1.60	0.94	1.37	1.10	0.86	6.36	**0.042**	0.08	1–3 *p* = 0.036
MaxROM F2	19.02	19.00	1.72	19.39	19.05	4.91	17.18	18.00	5.60	1.49	0.474	<0.01	
Hz F1	2.42	2.40	0.60	1.43	1.15	0.97	1.06	1.00	0.83	9.82	**0.007**	0.14	1–3 *p* = 0.007
MaxROM F1	13.98	10.00	6.89	12.36	14.50	6.32	7.74	7.00	4.84	8.52	**0.014**	0.11	2–3 *p* = 0.041
Grip strength (kg)	32.50	25.70	14.17	19.40	14.65	16.65	17.11	15.20	11.17	5.23	0.073	0.06	
Stabilized Hz wrist (cyc/s)	1.90	2.10	0.85	1.29	1.10	1.13	0.98	0.90	0.60	5.52	0.063	0.06	
MaxROM (mm)	20.42	20.20	1.32	19.08	19.70	6.62	20.42	21.30	8.69	0.24	0.889	<0.01	
Hz F5	2.50	3.00	0.97	1.65	1.70	0.97	1.54	1.30	0.91	4.31	0.116	0.04	
MaxROM F5	21.52	20.10	7.31	21.28	19.95	15.22	15.56	15.40	6.61	3.23	0.199	0.02	
Hz F4	2.54	3.00	0.94	1.68	1.70	0.99	1.49	1.20	0.93	5.04	0.080	0.05	
MaxROM f4	22.82	19.00	6.53	21.94	25.00	9.45	18.96	19.80	5.58	2.18	0.336	0.00	
Hz F3	2.50	3.00	0.97	1.68	1.70	0.99	1.54	1.30	0.91	4.43	0.109	0.04	
MaxROM F3	21.74	20.60	3.58	22.36	22.70	6.37	18.90	19.20	4.59	4.51	0.105	0.04	
Hz F2	2.52	3.00	0.98	1.68	1.70	0.99	1.54	1.30	0.91	4.43	0.109	0.04	
MaxROM F2	17.48	17.50	4.16	18.50	17.80	4.73	16.77	16.50	5.13	0.78	0.676	<0.01	
Hz F1	1.88	2.30	1.33	1.33	1.15	1.18	1.09	0.90	0.78	1.77	0.412	<0.01	
MaxROM F1	13.56	13.50	6.89	9.52	11.80	5.25	7.18	7.10	4.16	6.19	**0.045**	0.07	X
Grip strength (kg)	32.76	35.40	15.77	19.08	11.05	17.29	18.01	15.70	10.76	4.61	0.100	0.05	

Legend: 1—18–45 years old; 2—46-60 years old; 3—61 years and older; X—no significant differences between groups in post hoc analysis; ROM—range of motion.

**Table 4 ijerph-19-04703-t004:** Comparison of the results of parameters in the non-stabilized and stabilized position for the 18–45-year-old age group of people after stroke.

	NON-STABILIZED			STABILIZED					
Parameters	*M*	*Me*	*SD*	*M*	*Me*	*SD*	*Z*	*p*	*r*
Hz wrist (cyc/s)	2.16	2.40	0.51	1.90	2.10	0.85	−0.41	0.686	0.13
Wrist MaxROM (mm)	20.14	20.60	3.86	20.42	20.20	1.32	−0.27	0.786	0.09
Hz F5	2.40	2.40	0.60	2.50	3.00	0.97	−0.54	0.588	0.17
MaxROM F5	26.42	26.30	3.77	21.52	20.10	7.31	−1.48	0.138	0.47
Hz F4	2.40	2.40	0.60	2.54	3.00	0.94	−0.54	0.588	0.17
MaxROM F4	29.02	26.60	9.61	22.82	19.00	6.53	−0.94	0.345	0.30
Hz F3	2.40	2.40	0.60	2.50	3.00	0.97	−0.54	0.588	0.17
MaxROM F3	23.72	24.20	2.13	21.74	20.60	3.58	−0.37	0.715	0.12
Hz F2	2.40	2.40	0.60	2.52	3.00	0.98	−0.54	0.588	0.17
MaxROM F2	19.02	19.00	1.72	17.48	17.50	4.16	−0.94	0.345	0.30
Hz F1	2.42	2.40	0.60	1.88	2.30	1.33	−0.54	0.588	0.17
MaxROM F1	13.98	10.00	6.89	13.56	13.50	6.89	−0.41	0.686	0.13
Grip strength (kg)	32.50	25.70	14.17	32.76	35.40	15.77	0.00	1.000	0.00

Legend: M—mean; ROM—range of motion; SD—standard deviation.

**Table 5 ijerph-19-04703-t005:** Comparison of the results of parameters in the stabilized and stabilized position for the 46–60-year-old age group of people after stroke.

	NON-STABILIZED			STABILIZED					
Parameters	*M*	*Me*	*SD*	*M*	*Me*	*SD*	*Z*	*p*	*r*
Hz wrist (cyc/s)	1.15	0.90	0.90	1.29	1.10	1.13	−0.54	0.592	0.11
Wrist MaxROM (mm)	17.30	18.60	4.81	19.08	19.70	6.62	−1.65	0.099	0.34
Hz F5	1.58	1.60	0.94	1.65	1.70	0.97	−0.42	0.673	0.09
MaxROM F5	18.48	17.50	10.83	21.28	19.95	15.22	−0.71	0.480	0.14
Hz F4	1.58	1.60	0.94	1.68	1.70	0.99	−0.86	0.389	0.18
MaxROM F4	23.69	26.85	9.71	21.94	25.00	9.45	−2.87	**0.004**	0.59
Hz F3	1.58	1.60	0.94	1.68	1.70	0.99	−0.86	0.389	0.18
MaxROM F3	23.41	24.25	6.28	22.36	22.70	6.37	−1.69	0.091	0.34
Hz F2	1.58	1.60	0.94	1.68	1.70	0.99	−0.86	0.389	0.18
MaxROM F2	19.39	19.05	4.91	18.50	17.80	4.73	−1.49	0.135	0.30
Hz F1	1.43	1.15	0.97	1.33	1.15	1.18	−0.24	0.812	0.05
MaxROM F1	12.36	14.50	6.32	9.52	11.80	5.25	−2.28	**0.023**	0.46
Grip strength (kg)	19.40	14.65	16.65	19.08	11.05	17.29	−0.16	0.875	0.03

Legend: M—mean; ROM—range of motion; SD—standard deviation.

**Table 6 ijerph-19-04703-t006:** Comparison of the results of parameters in the stabilized and unstable position for the age group of 61 years and more after stroke.

	NON-STABILIZED			STABILIZED					
Parameters	*M*	*Me*	*SD*	*M*	*Me*	*SD*	*Z*	*p*	*r*
Hz wrist (cyc/s)	1.03	0.90	0.60	0.98	0.90	0.60	−0.69	0.489	0.07
Wrist MaxROM (mm)	15.50	14.90	8.51	20.42	21.30	8.69	−4.89	**<0.001**	0.53
Hz F5	1.39	1.10	0.87	1.54	1.30	0.91	−2.38	**0.018**	0.26
MaxROM F5	16.54	16.60	7.22	15.56	15.40	6.61	−1.49	0.136	0.16
Hz F4	1.37	1.10	0.88	1.49	1.20	0.93	−1.93	0.053	0.21
MaxROM F4	20.61	21.10	6..49	18.96	19.80	5.58	−2.30	**0.021**	0.25
Hz F3	1.37	1.10	0.87	1.54	1.30	0.91	−2.59	**0.010**	0.28
MaxROM F3	20.19	20.00	5.29	18.90	19.20	4.59	−2.24	**0.025**	0.24
Hz F2	1.37	1.10	0.86	1.54	1.30	0.91	−2.54	**0.011**	0.27
MaxROM F2	17.18	18.00	5.60	16.77	16.50	5.13	−0.69	0.492	0.07
Hz F1	1.06	1.00	0.83	1.09	0.90	0.78	−1.27	0.205	0.14
MaxROM F1	7.74	7.00	4.84	7.18	7.10	4.16	−0.73	0.468	0.08
Grip strength (kg)	17.11	15.20	11.17	18.01	15.70	10.76	−1.83	0.067	0.20

Legend: ROM—range of motion; M—mean; SD—standard deviation.

**Table 7 ijerph-19-04703-t007:** Comparison of age groups of the healthy subjects, in terms of the parameters measured in the unstable and stable.

Age Groups of the Healthy Subjects	18–45 Years (*n* = 5)			46–60 Years (*n* = 12)			61 Years and Older (*n* = 40)						
	*M*	*Me*	*SD*	*M*	*Me*	*SD*	*M*	*Me*	*SD*	*H(2)*	*p*	*η^2^*	*Post hoc*
Non-stabilized													
Hz wrist (cyc/s)	3.24	3.10	0.64	2.87	2.70	0.90	2.88	2.75	1.26	0.66	0.719	<0.01	
Wrist MaxROM (mm)	26.46	28.50	10.80	22.68	21.85	3.70	25.11	22.80	10.64	0.24	0.888	<0.01	
Hz F5	3.54	3.20	0.93	2.78	2.70	0.62	2.81	2.80	0.75	2.79	0.248	0.01	
MaxROM F5	16.60	13.60	8.03	22.84	23.30	5.11	20.05	19.65	9.31	3.50	0.174	0.03	
Hz F4	3.52	3.10	0.94	2.52	2.60	0.97	2.89	2.80	0.62	3.95	0.139	0.03	
MaxROM F4	19.42	18.00	6.41	22.00	21.70	5.44	23.80	22.25	10.70	0.67	0.714	<0.01	
Hz F3	3.54	3.20	0.93	2.78	2.70	0.62	2.88	2.90	0.76	2.75	0.252	0.01	
MaxROM F3	17.34	15.50	5.89	20.49	20.10	4.54	22.59	22.80	9.18	2.73	0.255	0.01	
Hz F2	3.54	3.20	0.93	2.77	2.70	0.61	2.95	2.90	0.62	3.02	0.221	0.02	
MaxROM F2	17.52	15.40	4.49	20.22	19.25	4.12	18.69	19.30	5.37	1.80	0.406	<0.01	
Hz F1	3.58	3.20	0.90	2.28	2.40	0.99	2.47	2.70	1.08	6.19	**0.045**	0.07	1–2 *p* = 0.043
MaxROM F1	10.50	10.20	4.63	14.79	16.40	5.94	14.03	12.70	5.76	2.23	0.329	0.00	
Grip strength (kg)	37.14	37.00	8.11	33.43	32.00	9.16	33.77	31.00	11.91	0.82	0.664	<0.01	
Stabilized													
Hz wrist (cyc/s)	3.28	3.50	0.69	3.25	3.40	0.71	2.62	2.50	0.80	8.01	**0.018**	0.11	2–3 *p* = 0.042
Wrist MaxROM (mm)	32.84	29.90	13.55	28.67	25.35	11.81	24.30	24.40	3.95	1.64	0.441	<0.01	
Hz F5	2.78	3.30	1.47	3.11	3.00	1.15	2.86	3.00	0.92	1.33	0.514	<0.01	
MaxROM F5	16.98	15.20	6.40	17.67	16.50	4.82	18.14	17.75	6.82	0.18	0.912	<0.01	
Hz F4	2.94	3.30	1.72	2.74	3.00	1.31	2.95	3.05	0.85	0.52	0.771	<0.01	
MaxROM F4	19.16	15.10	6.96	18.65	17.45	6.79	21.44	19.70	9.53	0.62	0.732	<0.01	
Hz F3	2.92	3.30	1.72	3.42	3.40	0.69	2.88	3.00	0.93	3.21	0.201	0.02	
MaxROM F3	19.36	16.80	5.00	17.74	16.45	6.23	20.41	21.70	7.01	3.06	0.216	0.02	
Hz F2	3.66	3.30	0.71	3.42	3.40	0.69	3.10	3.10	0.54	4.81	0.090	0.05	
MaxROM F2	18.86	18.60	5.16	17.12	16.40	3.67	19.05	19.85	4.90	2.53	0.282	0.01	
Hz F1	2.66	3.20	1.38	2.82	3.00	1.34	2.58	2.85	1.17	1.05	0.590	<0.01	
MaxROM F1	15.18	16.50	4.05	15.04	14.85	6.97	14.63	14.20	5.13	0.05	0.974	<0.01	
Grip strength (kg)	36.20	31.80	10.86	33.65	32.75	8.69	31.43	29.60	10.80	1.21	0.545	<0.01	

Legend: 1—18–45 years old; 2–46-60 years old; 3—61 years and older; X—no significant differences between groups in post hoc analysis; ROM—range of motion.

**Table 8 ijerph-19-04703-t008:** Comparison of the results of parameters in the unstable and stabilized position for the age group of 18–45 years of healthy people.

	NON-STABILIZED			STABILIZED					
Parameters	*M*	*Me*	*SD*	*M*	*Me*	*SD*	*Z*	*p*	*r*
Hz wrist (cyc/s)	3.24	3.10	0.64	3.28	3.50	0.69	−0.18	0.854	0.06
Wrist MaxROM (mm)	26.46	28.50	10.80	32.84	29.90	13.55	−2.02	**0.043**	0.64
Hz F5	3.54	3.20	0.93	2.78	3.30	1.47	−0.41	0.684	0.13
MaxROM F5	16.60	13.60	8.03	16.98	15.20	6.40	−0.41	0.684	0.13
Hz F4	3.52	3.10	0.94	2.94	3.30	1.72	−0.18	0.686	0.06
MaxROM F4	19.42	18.00	6.41	19.16	15.10	6.96	−0.67	0.854	0.21
Hz F3	3.54	3.20	0.93	2.92	3.30	1.72	−0.41	0.500	0.13
MaxROM F3	17.34	15.50	5.89	19.36	16.80	5.00	−1.48	0.686	0.47
Hz F2	3.54	3.20	0.93	3.66	3.30	0.71	−0.68	0.138	0.22
MaxROM F2	17.52	15.40	4.49	18.86	18.60	5.16	−0.94	0.496	0.30
Hz F1	3.58	3.20	0.90	2.66	3.20	1.38	−0.74	0.345	0.23
MaxROM F1	10.50	10.20	4.63	15.18	16.50	4.05	−1.75	0.461	0.55
Grip strength sitting (kg)	37.14	37.00	8.11	36.20	31.80	10.86	−0.41	0.080	0.13

Legend: M—mean; SD—standard deviation.

**Table 9 ijerph-19-04703-t009:** Comparison of the results of the parameters in the unstable and stabilized position for the age group of 46–60 healthy people.

	NON-STABILIZED			STABILIZED					
Parameters	*M*	*Me*	*SD*	*M*	*Me*	*SD*	*Z*	*p*	*r*
Hz wrist (cyc/s)	2.87	2.70	0.90	3.25	3.40	0.71	−1.74	0.083	0.36
Wrist MaxROM (mm)	22.68	21.85	3.70	28.67	25.35	11.81	−2.04	**0.041**	0.42
Hz F5	2.78	2.70	0.62	3.11	3.00	1.15	−2.12	**0.034**	0.43
MaxROM F5	22.84	23.30	5.11	17.67	16.50	4.82	−2.59	**0.034**	0.53
HzP4	2.52	2.60	0.97	2.74	3.00	1.31	−1.96	**0.010**	0.40
MaxROM F4	22.00	21.70	5.44	18.65	17.45	6.79	−2.20	**0.050**	0.45
Hz F3	2.78	2.70	0.62	3.42	3.40	0.69	−3.07	**0.028**	0.63
MaxROM F3	20.49	20.10	4.54	17.74	16.45	6.23	−1.26	**0.002**	0.26
Hz F2	2.77	2.70	0.61	3.42	3.40	0.69	−3.07	0.209	0.63
MaxROM F2	20.22	19.25	4.12	17.12	16.40	3.67	−2.12	**0.002**	0.43
Hz F1	2.28	2.40	0.99	2.82	3.00	1.34	−2.04	**0.034**	0.42
MaxROM F1	14.79	16.40	5.94	15.04	14.85	6.97	−0.16	**0.041**	0.03
Grip strength (kg)	33.43	32.00	9.16	33.65	32.75	8.69	−0.16	0.875	0.03

Legend: M—mean; SD—standard deviation; ROM—range of motion.

**Table 10 ijerph-19-04703-t010:** Comparison of the results of parameters in the stabilized and stabilized position for the age group of 61 and more healthy people.

	NON-STABILIZED			STABILIZED					
Parameters	*M*	*Me*	*SD*	*M*	*Me*	*SD*	*Z*	*p*	*r*
Hz wrist (cyc/s)	2.88	2.75	1.26	2.62	2.50	0.80	−1.05	0.294	0.12
Wrist MaxROM (mm)	25.11	22.80	10.64	24.30	24.40	3.95	−1.39	0.164	0.16
Hz F5	2.81	2.80	0.75	2.86	3.00	0.92	−1.41	0.159	0.16
MaxROM F5	20.05	19.65	9.31	18.14	17.75	6.82	−1.51	0.159	0.17
Hz F4	2.89	2.80	0.62	2.95	3.05	0.85	−1.55	0.132	0.17
MaxROM F4	23.80	22.25	10.70	21.44	19.70	9.53	−2.02	0.121	0.23
Hz F3	2.88	2.90	0.76	2.88	3.00	0.93	−0.70	**0.043**	0.08
MaxROM F3	22.59	22.80	9.18	20.41	21.70	7.01	−1.47	0.483	0.16
Hz F2	2.95	2.90	0.62	3.10	3.10	0.54	−1.49	0.143	0.17
MaxROM F2	18.69	19.30	5.37	19.05	19.85	4.90	−0.67	0.137	0.07
Hz F1	2.47	2.70	1.08	2.58	2.85	1.17	−1.38	0.502	0.15
MaxROM F1	14.03	12.70	5.76	14.63	14.20	5.13	−0.91	0.168	0.10
Grip strength (kg)	33.77	31.00	11.91	31.43	29.60	10.80	−1.63	0.364	0.18

Legend: M—mean; SD—standard deviation; ROM—range of motion.

## Data Availability

Data available on request from corresponding author.

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
