# Peer review of "Change in the Results of Motor Coordination and Handgrip Strength Depending on Age and Body Position—An Observational Study of Stroke Patients and Healthy Volunteers"

_ijerph, 2022, doi:10.3390/ijerph19084703_

Round 1
Reviewer 1 Report
In lines 47-49: "from the age of 18 to 45 we can observe the peak physical fitness. In people over forty, we can observe a slow decline in fitness and physical strength, some feel a lack of energy or fatigue and weakness, muscle mass and strength" - a bit inconsistent information regarding age.
In Lines 62 and 132 - please check punctuation marks
Author Response
Manuscript ID: ijerph-1634845
Type of manuscript: Article
Title: Change in the results of motor coordination and handgrip strength depending on age and body position - an observational study of stroke patients and healthy volunteers.
Dear Reviewers,
Thank you very much for the analysis of our manuscript. We really appreciate your comments and indication of fragments that should be corrected and explained. Considering your suggestions, all mistakes were corrected. In order to avoid misunderstandings, changes introduced in the text are marked in blue and additionally, the manuscript was sent in the change tracking mode.
Reviewer #1:
Thank you very much for the very quick and thorough analysis of our manuscript.
The following comments and answers:
In lines 47-49: "from the age of 18 to 45 we can observe the peak physical fitness. In people over forty, we can observe a slow decline in fitness and physical strength, some feel a lack of energy or fatigue and weakness, muscle mass and strength" - a bit inconsistent information regarding age.
Indeed, the information in lines 47 to 49 sounds inconsistent. The information on age ranges was taken from the World Health Organization's reports, but my message was not the best. Therefore, I slightly corrected the sentence so that the characteristics of the age group became clearer. The correction of the sentence is blue and visible in the track changes mode.
Thank you very much for pointing out this lack of consistency in the text.
Thank you very much for this comment.
In Lines 62 and 132 - please check punctuation marks
I checked and corrected the punctuation marks in lines 62 to 132. I also checked the entire text and I hope I made the appropriate linguistic corrections.
Thank you very much for your careful study of our manuscript.
Thank you very much for your time.
Reviewer 2 Report
This paper is an observational study on the result of motor coordination, depending on the age and position of the participants with and without stroke.
The subject is clearly introduced and the purpose of the study is well explained. The method is ideally described and the conclusion is supported by the results as expected.
As a minor comment, it is difficult to understand the main take-over of this study as compared to previous work. The `research value` section was probably added to circumvent this issue, but the authors could probably emphasize this aspect in this section and at the end of the introduction.
The other limitation, concerning the size of the groups, is clearly stated and explained in the limitation section. In addition, the authors could elaborate on the difference in physical fitness within the group of people with age > 60 (does it make sense to put in the same group people with age 60 and 80?). And also provide age standard deviation for each category, e.g. group with age > 60 (+- 5).
Author Response
Manuscript ID: ijerph-1634845
Type of manuscript: Article
Title: Change in the results of motor coordination and handgrip strength depending on age and body position - an observational study of stroke patients and healthy volunteers.
Dear Reviewers,
Thank you very much for the analysis of our manuscript. We really appreciate your comments and indication of fragments that should be corrected and explained. Considering your suggestions, all mistakes were corrected. In order to avoid misunderstandings, changes introduced in the text are marked in blue and additionally, the manuscript was sent in the change tracking mode.
Reviewer #2:
Thank you very much for the very quick and thorough analysis of our manuscript.
The following comments and answers:
Comments and Suggestions for Authors
This paper is an observational study on the result of motor coordination, depending on the age and position of the participants with and without stroke.
The subject is clearly introduced and the purpose of the study is well explained. The method is ideally described and the conclusion is supported by the results as expected.
Thank you very much for the good evaluation of our manuscript.
As a minor comment, it is difficult to understand the main take-over of this study as compared to previous work. The `research value` section was probably added to circumvent this issue, but the authors could probably emphasize this aspect in this section and at the end of the introduction.
I do not understand exactly what the reviewer meant when he wrote: "compared to the previous work". In the presented research, we have shown that there is a difference in the achieved results in different age groups. The oldest people achieve significant results in motor coordination, despite the fact that all groups are conducted in the same way, the examination procedure is the same. Compared to our other studies, in which we tested the solid position of the study (for example, with respect to the assessment of the dominant and non-dominant hand), the non-dominant one achieved significant results. In these studies, a stable position is important for the oldest patients, who also have a higher incidence of stroke. This shows that the use of a stable position makes a special sense, when we work to improve coordination and the handgrip strength, in the "weaker" patients like older, post-stroke, or had a non-dominant hand.
Taking into account the suggestions, I tried to make an explanation in the text. At the end of the introduction, I point out how difficult it is to improve the efficiency of the distal part of the upper limb and that strokes are more common in older people, which is why I made the correction, especially in the discussion section and under the research value sub-item. Moreover, in the discussion, I referred to our previous work, which resulted in the addition one of reference (position 40) to the list of references.
Thank you very much for this comment and suggestion.
The other limitation, concerning the size of the groups, is clearly stated and explained in the limitation section. In addition, the authors could elaborate on the difference in physical fitness within the group of people with age > 60 (does it make sense to put in the same group people with age 60 and 80?). And also provide age standard deviation for each category, e.g. group with age > 60 (+- 5).
In our research, we wanted to analyze which position is better, whether it is important for the improvement of motor coordination and handgrip strength, and whether, using the same test procedure, we will find differences in individual age groups in people after stroke and (for comparison) in healthy volunteers. We took into account the fact that, depending on age, the physical fitness of healthy people changes, and in the case of sick people (in this case after a stroke), the disease itself changes the ability to function and impairs patients more. People over 60 are more likely to experience strokes. Our oldest group is indeed the most numerous, as we explain in the study limitation. The age of the respondents was determined by the age of the patients in the Rehabilitation Department.
In the introduction to work, I developed information on the physical and intellectual fitness of people over 60 years of age. Moreover, I have presented the division into periods among people over 60 years of age, according to WHO.
On the other hand, data such as the average age of the respondents and the standard deviation were added and complete the characteristics of the respondents in Table 2.
Thank you very much for the very thorough analysis of our work and any comments.
Thank you very much for your time.
Reviewer 3 Report
The topic of this manuscript falls within the scope of International Journal of Environmental Research and Public Health. The topic of the manuscript is very interesting, relevant, and original.
It was an observational study. The aim of this study to assess the parameters of motor coordination and handshake strength in stroke patients and healthy volunteers in various age groups and selected torso and upper limb positions. The Authors concluded that passive stabilization of the trunk and upper limb improves motor coordination, especially in the oldest group of patients after stroke.
The strength of this paper: very interesting topic, introduction-relevant and concise; material and methods-the right choice of methodology methods, which was presented in comprehensible way; the obtained results are presented in the form of tables, which are clear and easy to understand; the discussion- supports the results properly and refers to the current literature in appropriate manner; the conclusions- based on the obtained results.
There are some comments in the reviewer opinion which should be taken under consideration by the Authors:
- in the discussion/limitation, please refer to the issue: frailty syndrome (cite: PMID: 31609229.)
- in the discussion, please underline those patients with stroke have also atherosclerosis in other vascular beds (PMID: 34110907.) and it may have influence on their rehabilitation results.
,
Author Response
Manuscript ID: ijerph-1634845
Type of manuscript: Article
Title: Change in the results of motor coordination and handgrip strength depending on age and body position - an observational study of stroke patients and healthy volunteers.
Dear Reviewers,
Thank you very much for the analysis of our manuscript. We really appreciate your comments and indication of fragments that should be corrected and explained. Considering your suggestions, all mistakes were corrected. In order to avoid misunderstandings, changes introduced in the text are marked in blue and additionally, the manuscript was sent in the change tracking mode.
Reviewer #3:
Thank you very much for the very quick and thorough analysis of our manuscript.
The following comments and answers:
Comments and Suggestions for Authors
The topic of this manuscript falls within the scope of International Journal of Environmental Research and Public Health. The topic of the manuscript is very interesting, relevant, and original.
It was an observational study. The aim of this study to assess the parameters of motor coordination and handshake strength in stroke patients and healthy volunteers in various age groups and selected torso and upper limb positions. The Authors concluded that passive stabilization of the trunk and upper limb improves motor coordination, especially in the oldest group of patients after stroke.
The strength of this paper: very interesting topic, introduction-relevant and concise; material and methods-the right choice of methodology methods, which was presented in comprehensible way; the obtained results are presented in the form of tables, which are clear and easy to understand; the discussion- supports the results properly and refers to the current literature in appropriate manner; the conclusions- based on the obtained results.
Thank you very much for the very good evaluation of our manuscript.
There are some comments in the reviewer opinion which should be taken under consideration by the Authors:
in the discussion/limitation, please refer to the issue: frailty syndrome (cite: PMID: 31609229.)
I referred in the section discussion sub-item study limitation to the above issue.
Thank you very much for this suggestion.
in the discussion, please underline those patients with stroke have also atherosclerosis in other vascular beds (PMID: 34110907.) and it may have an influence their rehabilitation results.
In the discussion and in the study limitation, I drew attention to the occurrence of atherosclerosis in other parts of the vascular bed in elderly people.
Thanks for paying attention to this. Thanks to your attentiveness, I have added more references to the work and to the reference list.
Thank you very much for the careful study of the manuscript.
Thank you very much for your time.
Reviewer 4 Report
The authors report an analysis of the results of an observational study. The study aimed to examine the relationship of the body position and the examined upper limb with motor coordination and handgrip strength in stroke patients and healthy people of various ages. The authors observed that significantly different results of motor coordination of the distal part of the upper limb could be obtained with and without passive stabilization of the trunk and shoulder, especially in older adults with stroke. These findings are interesting with potential therapeutic implications.
There are some comments.
Comments:
- MATERIAL AND METHODS (Line 192 on page 6): “the measurement of grip strength with a dynamometer was performed in both analyzed starting positions, after completing the range of motion and speed or frequency tests.” Were the participants allowed to rest before grip strength measurement? If yes, please state it. If not, a discussion of the possible effects on the grip strength results is suggested.
- MATERIAL AND METHODS (Table 1): The authors presented height, weight, and BMI in stroke patients and a healthy control group of different ages. Were there significant differences in these biometric data between stroke patients and the healthy control group? Also, were there significant differences in these biometric data between different age groups? Statistical tests are needed here. How would the differences, if any, relate to the differences in motor coordination? A discussion is recommended.
- MATERIAL AND METHODS (Table 2): Please present the following results in Table 2: trunk Control Test, FMA-UE motor function points, and Modified Ashworth Scale.
- MATERIAL AND METHODS (Table 2): Please report the following epidemiological data in different age groups: dominant hand, cerebral ischemic stroke unilateral subcortical, time post-stroke/episode, trunk control test, FMA-UE motor function points, and Modified Ashworth Scale.
- MATERIAL AND METHODS (Table 2): Were there significant differences in these epidemiological data between stroke patients and the healthy control group? Also, were there significant differences between different age groups? Statistical tests are needed. How would the differences, if any, relate to the differences in motor coordination? A discussion is recommended.
- There are numerous syntactical errors. The manuscript should be carefully reviewed by a person with expertise in writing in English.
Author Response
Manuscript ID: ijerph-1634845
Type of manuscript: Article
Title: Change in the results of motor coordination and handgrip strength depending on age and body position - an observational study of stroke patients and healthy volunteers.
Dear Reviewers,
Thank you very much for the analysis of our manuscript. We really appreciate your comments and indication of fragments that should be corrected and explained. Considering your suggestions, all mistakes were corrected. In order to avoid misunderstandings, changes introduced in the text are marked in blue and additionally, the manuscript was sent in the change tracking mode.
Reviewer #4:
Thank you very much for the very quick and thorough analysis of our manuscript.
Comments and Suggestions for Authors
The authors report an analysis of the results of an observational study. The study aimed to examine the relationship of the body position and the examined upper limb with motor coordination and handgrip strength in stroke patients and healthy people of various ages. The authors observed that significantly different results of motor coordination of the distal part of the upper limb could be obtained with and without passive stabilization of the trunk and shoulder, especially in older adults with stroke. These findings are interesting with potential therapeutic implications.
Thank you very much for such a good opinion about our work.
While improving the work, I took into account all your suggestions and comments.
The following comments and answers:
There are some comments.
Comments:
MATERIAL AND METHODS (Line 192 on page 6): “the measurement of grip strength with a dynamometer was performed in both analyzed starting positions, after completing the range of motion and speed or frequency tests.” Were the participants allowed to rest before grip strength measurement? If yes, please state it. If not, a discussion of the possible effects on the grip strength results is suggested.
Indeed, I did not describe the research methodology too precisely. In fact, I did not expect a break for rest during these tests, but in fact, each of the respondents had such a time (15 to 20 minutes) resulting from the procedure of preparation for the test. The break time was relatively short and maybe that is why it was not originally recorded at work. However, taking into account your important suggestion, I have made appropriate corrections to the text of the manuscript (Line 201 -203 on page 6). However, the text could be moved due to the corrections made. Track changes mode and the blue color of the font should make it easier to find the lines.
Moreover, I added appropriate study limitations.
Thank you very much for your insightful review of our work.
MATERIAL AND METHODS (Table 1): The authors presented height, weight, and BMI in stroke patients and a healthy control group of different ages. Were there significant differences in these biometric data between stroke patients and the healthy control group? Also, were there significant differences in these biometric data between different age groups? Statistical tests are needed here. How would the differences, if any, relate to the differences in motor coordination? A discussion is recommended.
In biometric data (height, weight, BMI), there were no significant differences between the post-stroke group and healthy volunteers as well as between the age groups. Due to the above, I decided that the results of the study will be attached as a supplement to the work (supplements 1 and 2).
Supplement 1. Assessment of differences in biometric data in the studied groups.
Supplement 2. Assessment of differences in biometric data in the studied age ranges.
I also made a corresponding note at work where I provide a reference to the supplement.
Thank you very much for pointing this out.
Thank you for this comment.
MATERIAL AND METHODS (Table 2): Please present the following results in Table 2: trunk Control Test, FMA-UE motor function points, and Modified Ashworth Scale.
As suggested, Table 2 presents all the indicated results for each age group. The change/addition of table 2 probably caused a shift of lines in the work, but the blue color of the font and the work in the change tracking mode make it easier to find the corrected fragments of the manuscript.
Thank you very much for this suggestion.
MATERIAL AND METHODS (Table 2): Please report the following epidemiological data in different age groups: dominant hand, cerebral ischemic stroke unilateral subcortical, time post-stroke/episode, trunk control test, FMA-UE motor function points, and Modified Ashworth Scale.
As suggested, all epidemiological data such as dominant hand, unilateral subcortical ischemic stroke, time after stroke/episode, torso control test, FMA-UE motor function points, and Modified Ashworth Scale are given for each age group. The blue font underlines the correction in table 2.
Thank you very much for this comment.
MATERIAL AND METHODS (Table 2): Were there significant differences in these epidemiological data between stroke patients and the healthy control group? Also, were there significant differences between different age groups? Statistical tests are needed. How would the differences, if any, relate to the differences in motor coordination? A discussion is recommended.
Of course, there are significant differences between the stroke group and healthy volunteers. Healthy volunteers constituted the control group, which was to consist of healthy and fit individuals. After all, the aim of the study was to assess the importance of the body position and age of the examined people on the results of coordination and handgrip strength.
Therefore, I decided that the calculations for healthy people with the same data (TCT 100, FMA UE 66, MAS 0) do not make sense.
However, there were no significant differences between the studied age groups of patients after stroke (Supplements 3, 4, and 5):
Supplement 3. Assessment of differences in TCT and FMA UE results in the studied age ranges of patients after stroke.
Supplement 4. Assessment of differences in MAS results in the studied age ranges of patients after stroke.
Supplement 5. There are no differences between age groups among patients after stroke in terms of the following parameters: TCT, FMA UE, MAS.
- TCT: H = 2,56; p = 0,278; η2 = 0,01
- FMA-UE: H = 0,70; p = 0,704; η2 < 0,01
- MAS p = 1,000; V = 0,04
There is a relevant comment in the text and an added supplement 3, 4, and 5, in the form of tables.
There are numerous syntactical errors. The manuscript should be carefully reviewed by a person with expertise in writing in English.
The entire manuscript has been checked and corrected for syntax, punctuation, and the use of English words. I checked the manuscript for grammar and typos. I hope I have taken into account all the mistakes that exist. Thank you very much for this comment.
Thank you for your very thorough analysis of our manuscript.
Thank you very much for your time.
Round 2
Reviewer 4 Report
The authors have addressed all the concerns in this version of the manuscript.